# Integrative Transcriptome and Metabolome Analysis Reveals Candidate Genes Related to Terpenoid Synthesis in *Amylostereum areolatum* (*Russulales: Amylostereaceae*)

**DOI:** 10.3390/jof11050383

**Published:** 2025-05-16

**Authors:** Lixia Wang, Ningning Fu, Ming Wang, Zhongyi Zhan, Youqing Luo, Jianrong Wu, Lili Ren

**Affiliations:** 1Key Laboratory of Forest Disaster Warning and Control of Yunnan Province, Southwest Forestry University, Kunming 650224, China; wlxynl@163.com (L.W.); 18600508400@163.com (Z.Z.); 2Beijing Key Laboratory for Forest Pest Control, Beijing Forestry University, Beijing 100083, China; funingning2012@sina.com (N.F.); 13020028768@163.com (M.W.); youqingluo@126.com (Y.L.); 3College of Forestry, Hebei Agricultural University, Baoding 071000, China; 4Guangdong Key Laboratory of Animal Conservation and Resource Utilization, Institute of Zoology, Guangdong Academy of Sciences, Guangzhou 510260, China; 5Guangdong Public Laboratory of Wild Animal Conservation and Utilization, Institute of Zoology, Guangdong Academy of Sciences, Guangzhou 510260, China; 6Sino-French Joint Laboratory for Invasive Forest Pests in Eurasia, INRAE-Beijing Forestry University, Beijing 100083, China

**Keywords:** *Sirex noctilio*, *Amylostereum areolatum*, metabolome, transcriptome, metabolome analysis, terpenoids

## Abstract

*Amylostereum areolatum* (Chaillet ex Fr.) Boidin (*Russulales: Amylostereaceae*) is a symbiotic fungus of *Sirex noctilio* Fabricius that has ecological significance. Terpenoids are key mediators in fungal–insect interactions, yet the biosynthetic mechanisms of terpenoids in this species remain unclear. Under nutritional conditions that mimic natural growth, *A. areolatum* was sampled during the lag phase (day 7), exponential phase (day 14), and stationary phase (day 21). Metabolome (solid-phase microextraction (SPME) combined with gas chromatography–mass spectrometry (GC-MS) and liquid chromatography–mass spectrometry (LC-MS)) and transcriptome (Illumina NovaSeq) profiles were integrated to investigate terpenoid–gene correlations. This analysis identified 103 terpenoids in *A. areolatum*, substantially expanding the known repertoire of terpenoid compounds in this species. Total terpenoid abundance progressively increased across three developmental stages, with triterpenoids and sesquiterpenoids demonstrating the highest diversity and abundance levels. Transcriptomic profiling (61.66 Gb clean data) revealed 26 terpenoid biosynthesis-associated genes, establishing a comprehensive transcriptional framework for fungal terpenoid metabolism. Among 11 differentially expressed genes (DEGs) (|log2Fold Change| ≥ 1, adjusted *p* < 0.05), *HMGS1*, *HMGR2*, and *AaTPS1-3* emerged as key regulators potentially governing terpenoid biosynthesis. These findings provide foundational insights into the molecular mechanisms underlying terpenoid production in *A. areolatum* and related basidiomycetes.

## 1. Introduction

*Sirex noctilio* is a globally significant quarantine pest that is native to Europe and North Africa. It has caused severe damage in various countries and regions after invasion. Host trees infested by *S. noctilio* exhibit needle blight and discoloration, leading to severe decline and mortality, resulting in substantial economic losses [1,2]. In China, *S. noctilio* was first detected in Heilongjiang Province in 2013, where it primarily attacks weakened Mongolian Scots pines (*Pinus sylvestris* var. *mongolica*) in field conditions [3,4]. Currently, the distribution of *S. noctilio* in China has expanded to 22 locations, including the three northeastern provinces and Inner Mongolia [5,6]. *Amylostereum areolatum* (Chaillet ex Fr.) Boidin (*Basidiomycotina: Amylostereaceae*), a fungal symbiont associated with *S. noctilio*, exhibits an obligate mutualistic relationship with the wood wasp, characterized by strict dependency [3]. The symbiotic fungi are stored in the fold between the first and second abdominal segments of *S. noctilio* larvae and in the mycangium of female adults [7,8]. During oviposition, *S. noctilio* directly inoculates its fungal symbiont into host trees, where it synergizes with wasp-specific phytotoxins to disrupt the host’s physiological metabolism and compromise structural integrity. This dual pathogenesis progressively weakens the tree’s defensive capacity and accelerates physiological decline, ultimately creating an optimized environment that attracts conspecific females for subsequent oviposition events [9,10,11].

Volatile-mediated host selection mechanisms have been extensively documented in this symbiotic system. Fernández Ajó et al. demonstrated that both host-derived volatiles and fungal emissions from *A. areolatum* function as olfactory cues for *S. noctilio*, with fungal volatiles exhibiting significantly greater attraction efficacy [12]. GC-MS analyses identified key semiochemicals in *A. areolatum* emissions, including 1-(1,5-dimethyl-4-hexenyl)-4-methylbenzene, 4-methoxybenzaldehyde, 6-methyl-5-hepten-2-one, trans-3-hexenyl acetate, linalool, and geraniol, all showing dose-dependent electrophysiological responses in *S. noctilio* antennae [13,14]. In addition, Masagué et al. systematically conducted volatile profiles studies of *A. areolatum* cultured on artificial media, identifying 62 bioactive compounds that elicited antennal responses in female wasps through gas chromatography–electroantennographic detection (GC-EAD) [15]. Our research group has further validated the chemoecological attraction between symbiotic fungi and their hosts through comprehensive behavioral assays. Both laboratory-based bioassays (‘Y’-tube olfactometer and wind tunnel experiments) and field trials have consistently demonstrated *A. areolatum*’s superior attraction capacity compared to other fungal associates (*A. chailletii* and endophytic competitors) [16,17].

Numerous studies have demonstrated the critical role of terpenoids in insect host selection. Host plant-derived terpenes, particularly α-pinene and β-pinene, have been identified as essential chemical attractants for *S. noctilio*. Bashford’s research established that a 7:3 ratio of α-pinene to β-pinene exhibited optimal attraction efficacy for *S. noctilio* [18]. Concurrently, Böröczky et al. characterized (Z)-7-heptacosene, (Z)-7-nonacosene, and (Z)-9-nonacosene as female-specific sex pheromones in this species [19]. Among the bioactive volatile compounds emitted by symbiotic fungi that attract *S. noctilio*, multiple terpenoids have been identified, including 1-(1,5-dimethyl-4-hexenyl)-4-methylbenzene, 2-carene, piperitone, 2,2,8-trimethyltricyclo[6.2.2.01,6]dodec-5-ene, beta-bisabolol, terpene, and alloaromadendrene [13,15,16,17]. Nevertheless, despite these discoveries, the complete terpenoid profile of *A. areolatum* and its associated biosynthetic mechanisms remain unelucidated.

Terpenoids, defined as oxygenated or chemically modified derivatives of terpenes primarily comprising alcohols, aldehydes, ketones, and esters, represent the largest group of natural products in fungi [20]. Basidiomycetes exhibit a rich diversity of sesquiterpenoids, diterpenoids, and triterpenoids compared to ascomycetes [21]. It is well known that terpene synthases play an important role in the process of terpenoid synthesis. Terpenoids in fungi are predominantly synthesized via the mevalonate (MVA) pathway. To date, numerous terpene synthases (TPS) in fungi have been functionally validated, with sesquiterpene synthases (STS) constituting the most predominant class and monoterpene synthases (MTS) representing the least represented group [22,23,24]. Furthermore, integrated analysis of terpenoid-associated genes and metabolites has been conducted in *Sanghuangporus baumii* [25], *Ganoderma lucidum* [26], *Antrodia cinnamomea* [27], *Taiwanofungus gaoligongensis* [28], and *Hericium coralloides* [29], elucidating the key enzymes within their terpenoid biosynthetic pathways. Notably, although Fu et al. have sequenced the genome of *A. areolatum* and annotated *terpene synthase (TPS)* genes, the metabolic pathways of terpenoids in the symbiotic fungus associated with *S. noctilio* have yet to be investigated [30].

In this study, we investigated terpenoid metabolism in *A. areolatum* through integrated transcriptomic and metabolomic analyses of mycelial samples collected at three developmental stages (7, 14, and 21 days). Our objectives were to (1) characterize stage-specific terpenoid accumulation patterns, (2) map gene expression profiles associated with terpenoid biosynthetic pathways, and (3) identify key regulatory genes and uncover the mechanisms driving terpenoid metabolic divergence across fungal growth phases. This work provides a framework for understanding how basidiomycete fungi coordinate secondary metabolism during developmental transitions.

## 2. Materials and Methods

### 2.1. Strains and Growth Conditions

*Amylostereum areolatum* was isolated from the female wood wasps collected from the Xindian Forest Farm, Heilongjiang Province, China. The strain was identified and preserved in the Beijing Key Laboratory for the Control of Forest Pest, Beijing Forestry University, Beijing, China. We formulated the symbiotic fungal culture medium to closely mimic natural ecological conditions while satisfying experimental nutritional requirements. The formulation contained damaged Mongolian Scots pine wood powder (175.0 g) (taken from the edge of *S. noctilio* burrow), malt extract (3.0 g), yeast extract (0.3 g), agar (15.0 g), and distilled water (1000 mL), which was sterilized at 117 °C for 20 min [31]. After purification and identification, the fungi were inoculated into the culture medium and incubated at 25 °C in the dark. We collected mycelia on days 7, 14, and 21 (Figure 1A), referring to the experimental study by Sarvary, Jofré, et al. [13,32]. Based on fungal growth rate determination (Figure 1B), the 7-, 14-, and 21-day-old cultures were classified into the lag phase, exponential phase, and stationary phase of *A. areolatum* growth, respectively. There were three replicates for each of the three stages of fungi, which were used for terpenoid metabolic profiling and transcriptome analysis. For metabolite profiling, mycelial samples (50 mg fresh weight) were subjected to extraction protocols in parallel with sawdust-based medium and Potato Dextrose Agar (PDA) matrix blanks to normalize substrate-induced spectral interference. All samples were immediately frozen in liquid nitrogen and stored at −80 °C until required.

### 2.2. Detection and Analysis of Metabolites

Gas chromatography–mass spectrometry (GC-MS) is highly effective for analyzing volatile compounds. In contrast, liquid chromatography–mass spectrometry (LC-MS) is particularly suitable for detecting high-molecular weight and thermally unstable compounds. By combining these two techniques, a comprehensive analysis of the metabolites of *A. areolatum* can be achieved.

LC-MS metabolite extraction was initiated with precisely weighed 50 mg mycelial aliquots combined with 400 μL of methanol/water (4:1, *v*/*v*) containing 0.02 mg/mL of L-2-chloro-phenylalanine internal standard. Pre-chilled homogenization at −10 °C (50 Hz, 6 min) preceded the following sequential extraction phases: primary ultrasonication at 5 °C (40 kHz, 30 min), followed by −20 °C static incubation (30 min). Centrifugation at 13,000× *g* (4 °C, 15 min) yielded supernatants, 20 μL of which were pooled as QC samples. Residual supernatants were transferred to autosampler vials for instrumental analysis. The metabolites were separated chromatographically using a Thermo UHPLC system, which was fitted with an ACQUITY UPLC HSS T3 column (dimensions: 100 mm × 2.1 mm; internal diameter particle size: 1.8 μm; Waters, Milford, MA, USA). The sample injection volume was 2 μL, and the column temperature was maintained at 40 °C. Mobile phase A is 95% water and 5% acetonitrile containing 0.1% formic acid, and mobile phase B is 47.5% isopropanol, 47.5% acetonitrile, and 5% water containing 0.1% formic acid. The flow rate of the solvent is 0.40 mL/min. The mobile phase elution gradient is set as follows: 0–0.1 min, 100%A; 0.1–2 min, 95%A + 5%B; 2–9 min, 75%A + 25%B; 9–13 min, 100%B; 13–13.1 min, 100%B, 13.1–16 min, 100%A. Mass spectrometry analyses were conducted using a Thermo UHPLC-Q Exactive Mass Spectrometer system coupled with an electrospray ionization source, which was operated in both positive and negative ion modes. The mass spectrometric parameters are set as follows: the spray voltage is 3500 v in positive ion mode and −2880 v in negative ion mode; the heating temperature is 400 °C; the capillary temperature is 320 °C; the sheath gas flow rate is 40arb; the auxiliary air velocity is 10arb; and the scanning range is 70–1050 *m*/*z*. Raw LC-MS data were preprocessed using Progenesis QI v2.4 (Waters Corporation, Milford, CT, USA) according to the following steps: baseline correction, peak detection (signal-to-noise ratio ≥ 3), retention time alignment, and peak area integration. This generated a data matrix containing retention time (RT), mass-to-charge ratio (*m*/*z*), and intensity values. Putative metabolite annotation was performed by matching the data against the following public databases: HMDB (http://www.hmdb.ca/, accessed on 28 August 2021), Metlin (https://metlin.scripps.edu/, accessed on 28 August 2021), GNPS (https://gnps.ucsd.edu/ProteoSAFe/static/gnps-splash.jsp, accessed on 28 August 2021), and Majorbio databases. Missing values were imputed with the minimum positive value detected in the dataset to minimize bias introduced by zero replacement during log transformation. Data normalization was performed using total sum normalization to correct for batch effects. Features were retained only if they were detected in all three biological replicates (n = 3) of at least one experimental group. At the same time, variables with relative standard deviation (RSD) > 30% of QC samples were removed, and log10 logarithmization was performed to obtain the final data matrix for subsequent analysis.

For the determination of volatile compounds, we utilized a 50/30 μm DVB/CAR on PDMS fiber, along with a 7890B-5977B GC-MS system equipped with a DB-Wax chromatographic column (30 m × 0.25 mm × 0.25 µm, Agilent, Santa Clara, CA, USA). The 50 mg mycelium sample and 2 uL of internal standard (2-Octanol,(S)-, 50 μg/mL) were placed in a 20 mL headspace vial, sealed, incubated at 250 rpm for 15 min, and adsorbed at 50 °C for 30 min using the extraction fiber, followed by thermal desorption in the GC sampler for 5 min. The interface and injection temperatures were maintained at 260 °C. The column temperature gradient was programmed as follows: an initial hold at 40 °C for 5 min, followed by a temperature increase of 5 °C per minute to 220 °C and then a rapid increase of 20 °C per minute to 250 °C, which was maintained for 2.5 min. Instrument parameters were optimized with the quadrupole and ion source thermally stabilized at 150 °C and 230 °C, respectively. Ultra-high-purity helium (99.999%) served as the carrier gas throughout the analytical sequence. The ionization mode employed was electron ionization (EI), and full scan mode was utilized for data acquisition. The chromatograms and mass spectra of the volatile compounds from the symbiotic fungi were analyzed, and the characteristic peaks were identified using the NIST 2017 database to obtain the original data of the volatiles. After obtaining raw metabolite data, the relative quantification of metabolite concentration was calculated using the internal standard normalization method in the data preprocessing stage. The data shown are the average of three replicates.

To elucidate stage-specific terpenoid variations in *A. areolatum*, orthogonal partial least squares–discriminant analysis (OPLS-DA) was implemented through the MetaboAnalyst package in R (3.5.1) for discriminatory metabolite identification. Significant differences produced in the terpenoids were subsequently filtered using combined thresholds of VIP >1 and a fold change of ≥2 or a fold change of ≤0.5, thereby ensuring metabolic feature selection robustness.

### 2.3. RNA Extraction and Transcriptome Data Analysis

RNA extraction from *A. areolatum* was performed using the EZgene™ Fungal RNA Kit (Biomiga, San Diego, CA, USA), followed by comprehensive quality assessment procedures. The NanoDrop 2000 spectrophotometer quantified RNA concentration and purity, while electrophoretic separation on 1% agarose gels verified nucleic acid integrity. RNA Integrity Number (RIN) validation was subsequently conducted using the Agilent 2100 Bioanalyzer system. Shanghai Majorbio Technology Co. Ltd. (Shanghai, China) was commissioned to perform 2 × 150 bp transcriptome sequencing on the Illumina Novaseq 6000 sequencing platform.

First, the software SeqPrep (https://github.com/jstjohn/SeqPrep, accessed on 18 October 2021) and Sickle (https://github.com/najoshi/sickle, accessed on 18 October 2021) were used for raw data quality control, and then HISSAT2 was used to compare the raw clean data (reads) with the *A. areolatum* reference genome (INSDC: SAXG00000000.1; BioProject: PRJNA513942) to obtain mapped data (reads) [30]. Processed reads are deposited in NCBI’s SRA under BioProject PRJNA1162567. Quantitative analysis of gene expression levels was performed using RSEM Version 1.3.3, and gene expression levels were measured using the number of transcripts read per million (TPM). DESeq2 version 1.42.0 was used to analyze the original count data and to screen for DEGs, and genes satisfying |log2Fold Change| ≥ 1 and *p*-adjust < 0.05 were defined as differentially expressed genes (DEGs). Functional annotation leveraged seven databases: NR (NCBI non-redundant protein sequences database, https://www.ncbi.nlm.nih.gov/protein, accessed on 18 October 2021), PFAM (http://pfam.xfam.org/, accessed on 18 October 2021), GO (Gene Ontology, http://www.geneontology.org, accessed on 18 October 2021), KEGG (Kyoto Encyclopedia of Genes and Genomes, http://www.genome.jp/kegg/, accessed on 18 October 2021), EggNOG (http://eggnogdb.embl.de/#/app/home, accessed on 18 October 2021), and Swiss-Prot (http://web.expasy.org/docs/swiss-prot_guideline.html, accessed on 18 October 2021).

### 2.4. Comprehensive Analysis of the Metabolome and Transcriptome

Pearson correlation analysis was performed between sesquiterpenoids, and their associated differentially expressed genes (DEGs) were analyzed based on metabolite abundance and gene expression profiles using the psych package in R (v4.4.3). A Pearson correlation coefficient greater than 0.8 served as a threshold high correlation [26,34,35].

### 2.5. Real-Time Quantitative PCR

A PrimeScript^TM^ RT reagent Kit with gDNA Eraser (No.RR047A, Takara, Dalian, China) was used for reverse transcription–polymerase chain reaction, RT-PCR, and the specific methods were used according to the instructions. Specific primers for the target gene were designed using IDT (Integrated DNA Technologies|IDT, idtdna.com, accessed on 18 October 2021) and Primer 3.0 (https://bioinfo.ut.ee/primer3-0.4.0/, accessed on 18 October 2021) (*Gene_EVM004641*) and synthesized using RuiBiotech (Beijing, China). The products ranged in length from 80 to 200 bp, with the reference gene P450 (cytochrome P450) selected for comparison [36]. Real-time quantitative polymerase chain reaction amplification (RT-qPCR) was performed using the TB Green^®^ Premix Ex Taq™ II (Takara, Osaka, Japan) kit. The reaction system and amplification procedures were performed according to the experimental instructions. Relative gene expression was calculated using the 2^−ΔΔCt^ method with a Bio-Rad CFX96 PCR system (Hercules, CA, USA) in three technical replicates.

## 3. Results

### 3.1. Analysis of Terpenoid Components in A. areolatum

To comprehensively characterize the metabolic dynamics of *A. areolatum*, we acquired metabolite profiles at 7, 14, and 21 days using LC-MS and GC-MS (Appendix A). Seven hundred and sixty-two metabolites were obtained from the LC-MS metabolic set of *A. areolatum* cultured in the damaged Mongolian Scots pine wood powder medium, which contains 90 terpenoids, accounting for 14.78% of all the metabolites annotated in HMDB (609). Among these compounds, there are 8 monoterpenes, 23 sesquiterpenes, 9 diterpenes, 6 sesquiterpenes, and 44 triterpenes (Appendix A). As shown in Figure 2, the highest terpenoid abundance was detected in fungal samples cultured for 21 days. As further evidenced by the clustering heatmap of total terpenoids, 70% of the terpenoid compounds exhibited their peak relative content in the 21-day cultured samples (Appendix A). Notably, triterpenoids consistently exhibited the highest total abundance across all growth stages, followed by sesquiterpenoids, regardless of the cultivation phase (Figure 2A). A total of 57 volatile compounds were obtained via SPME extraction and GC-MS separation and identification, including 6 monoterpenes (10.53%) and 7 sesquiterpenoids (12.28%) (Appendix A). In contrast to the terpenoid profiles detected using LC-MS, 61.54% of volatile terpenoids demonstrated their maximum detection abundance in 7-day cultured samples (Figure 2B).

In the analysis of terpenoid metabolites detected using LC-MS, the OPLS-DA models for S14 vs. S7, S21 vs. S14, and S21 vs. S7 exhibited R^2^X, R^2^Y, and Q^2^ values of 0.833, 0.993, and 0.98; 0.683, 0.926, and 0.847; and 0.816, 0.995, and 0.976, respectively (Appendix A). All these indices exceeded 0.5, indicating that the analytical models were stable and reliable for subsequent analysis. The score plots of the models revealed a clear separation between samples from the two time periods (Appendix A). Similarly, in the analysis of terpenoid metabolites detected using GC-MS, the OPLS-DA models for S14 vs. S7, S21 vs. S14, and S21 vs. S7 demonstrated R^2^X, R^2^Y, and Q^2^ values of 0.881, 0.997, and 0.981; 0.773, 0.997, and 0.98; and 0.947, 0.997, and 0.988, respectively (Appendix A). All indices were greater than 0.5, confirming the stability and reliability of the models for further analysis. The score plots also showed a significant separation between samples from the two time periods (Appendix A). Differential metabolite analysis revealed that the numbers of differential terpenoids in three groups, namely, S14 vs. S7, S21 vs. S14, and S21 vs. S7, were 40, 37, and 54, respectively (Appendix A). Among these differential metabolites, 11 terpenoids, including (Z)-alpha-bergamotenoic acid, helinorbisabone, 4,5-dihydrovomifoliol, 2-O-protocatechuoylalphitolic acid, 22-acetylpriverogenin B, melilotigenin, lucidenic acid K, (17alpha,23S)-Epoxy-28,29-dihydroxy-27-norlanost-8-ene-3,24-dione, ganoderic acid A, 5,9,11-trihydroxyprosta-6E,14Z-dien-1-oate, and dihydrocarvone, exhibited significant differences across all three groups (Figure 2C). Among the 30 sesquiterpenes analyzed, 22 differentially metabolized sesquiterpenes were detected in one or more comparative groups. Additionally, among the terpenoids detected using LC-MS, a higher proportion showed an upregulation trend in different comparison groups: S14 vs. S7 (28/33), S21 vs. S14 (29/32), and S21 vs. S7 (40/45). Conversely, among the terpenoids detected by GC-MS, a higher proportion showed a downregulation trend in different comparison groups: S14 vs. S7 (5/7), S21 vs. S14 (4/5), and S21 vs. S7 (7/9) (Appendix A).

### 3.2. RNA-Seq and Transcriptomic Assembly

To identify genes involved in terpenoid biosynthesis in *A. areolatum*, nine qualified total RNA samples were sequenced. After conducting transcriptome data quality control, 61.66 Gb of clean data were obtained, with each sample reaching 6.85 Gb or higher and the GC content ranging from 57.19% to 57.54%. The average error rate of sequencing bases was less than 0.1%, Q20 was higher than 97.81%, and Q30 was higher than 93.98%. The minimum comparison rate between the clean reads of each sequencing sample and the reference genome is 87.81% (Table 1). The results indicate that the quality of the transcriptome data measured is qualified and can be used for subsequent analysis.

### 3.3. Gene Annotation and Functional Classification

In this reference genome-guided transcriptome study, all analyzed genes were derived from the annotated loci of the reference genome. Expressed genes were defined as those exhibiting detectable transcriptional activity in at least one biological sample, while non-expressed genes showed zero read counts across all samples. A total of 15,654 genes were obtained via transcriptome sequencing, including 11,894 expressed genes. Six databases (NR, PFAM, GO, KEGG, EggNOG, and Swiss-Prot) provided functional annotations for 11,002 genes (Figure 3).

Functional annotation revealed 8804 sequence-homologous genes systematically classified into three ontologies with 45 GO terms in the GO database. Specifically, 7221 (82.02%) genes were assigned to molecular function (MF), 5374 (61.04%) were assigned to cellular component (CC), and 4212 (47.84%) were assigned to biological process (BP). In other words, a greater proportion of the expressed genes were associated with molecular functions (Figure 4A). At the same time, among the 17 GO terms annotated to biological processes, 2924 (24.58%) genes were involved in the metabolic process (GO:0008152).

KEGG pathway annotation mapped 1824 genes to 122 metabolic modules, with translation (388 genes), carbohydrate metabolism (367 genes), and transport and catabolism (334 genes) emerging as the top three enriched pathways (Figure 4B). In addition, 26 genes were mapped to the ‘Metabolism of terpenoids and polyketides’, including sesquiterpenoid and triterpenoid biosynthesis (ko00909, 3 genes, *Gene_EVM009593*, *Gene_EVM008329*, *Gene_EVM006973*) and terpenoid backbone biosynthesis (ko00900, 23 genes, *Gene_EVM012320*, *Gene_EVM011243*, *Gene_EVM002105*, *Gene_EVM000360*, *Gene_EVM012186*, *Gene_EVM004075*, *Gene_EVM003694*, *Gene_EVM009139*, *Gene_EVM007424*, *Gene_EVM000380*, *Gene_EVM007998*, *Gene_EVM007970*, *Gene_EVM007969*, *Gene_EVM001820*, *Gene_EVM007966*, *Gene_EVM001819*, *Gene_EVM005538*, *Gene_EVM002515*, *Gene_EVM002681*, *Gene_EVM010953*, *Gene_EVM010955*, *Gene_EVM000972*, and *Gene_EVM009451*) (Appendix A).

Additionally, seven terpenoid biosynthesis-related genes (*Gene_EVM002513*, *Gene_EVM003164*, *Gene_EVM004641*, *Gene_EVM007245*, *Gene_EVM006403*, *Gene_EVM006404*, and *Gene_EVM003163*) were identified through functional annotation using the NR, Swiss-Prot, and GO databases, supplemented by BLAST-P homology searches (https://blast.ncbi.nlm.nih.gov/Blast.cgi, accessed on 2 February 2022) against the NR database (Appendix A). The phylogenetic tree (Figure 4C) indicates that the genes *Gene_EVM006403*, *Gene_EVM006404*, *Gene_EVM002513*, *Gene_EVM003164*, *Gene_EVM004641*, and *Gene_EVM007245* may encode sesquiterpene synthases. We have designated these genes as *AaTPS1* through *AaTPS6*, respectively.

### 3.4. Identification of Differentially Expressed Genes (DEGs)

To fully explore potential DEGs in the fungal growth stage, 1762 genes were identified by comparing the three groups (S14 vs. S7, S21 vs. S14, and S21 vs. S7). The longest sampling interval group (S21 vs. S7) exhibited the highest number of DEGs (n = 1325), with 602 upregulated and 723 downregulated genes. Comparative analysis of other groups revealed 402 DEGs (166 upregulated, 236 downregulated) between S14 and S7, and 860 DEGs (383 upregulated, 477 downregulated) when comparing S21 vs. S14. All DEGs underwent k-means clustering analysis, categorizing 1762 genes into six distinct expression clusters (116–467 genes per cluster; Figure 5). Genes within each cluster exhibited concordant expression patterns, suggesting their potential synergistic roles during the same temporal phase. In the six gene clusters analyzed in this study, the expression levels of 116 genes in Subclass 3 initially decreased before subsequently increasing. Meanwhile, the expression levels of 467 genes in Subclass 5 demonstrated a continuous upward trend.

### 3.5. Functional Classification of DEGs

Functional enrichment analysis identified the top 30 GO terms ranked by statistical significance (lowest adjusted *p*-values), with their enrichment magnitudes visualized in a bar graph (Figure 6A). Among the 23 GO terms of molecular function (MF), the largest related term was ‘oxidoreductase activity’. Differentially expressed genes (DEGs) within the cellular component ontology were predominantly localized to membrane-associated structures, specifically integral (GO:0016021) and intrinsic (GO:0031224) membrane components. In the category of biological process (BP), ‘small molecule catabolic process’ was the largest related term.

To further investigate the potential functions of these DEGs, KEGG pathway enrichment analyses were performed on them. According to the KEGG pathway enrichment analysis, 260 genes were significantly enriched into 99 pathways, of which MAPK signaling pathway—yeast (17), peroxisome (15), and amino sugar and nucleotide sugar metabolism (15) were the three richest pathways. A total of 167 DEGs are concentrated in 65 pathways called the ‘Metabolism’ category, which is the highest enrichment classification (Figure 6B). Among these, five genes were enriched to ‘Terpenoid backbone biosynthesis’ (map00900, *Gene_EVM007969*, *Gene_EVM002681*, *Gene_EVM002515*, *Gene_EVM010955*, and *Gene_EVM007970*), and one gene was enriched to ‘Sesquiterpenoid and triterpenoid biosynthesis’ (map00909, *Gene_EVM008329*) (Table 2). In addition, we also found five terpene synthase genes in DEGs (*Gene_EVM002513*, *Gene_EVM003164*, *Gene_EVM006403*, *Gene_EVM006404*, and *Gene_EVM007245*) (Table 2).

### 3.6. Genes Involved in Terpenoid Biosythesis

To elucidate the underlying mechanisms governing terpenoid accumulation patterns during *A. areolatum* developmental stages, transcriptional profiling of biosynthesis pathway genes was conducted. Analysis revealed 26 terpenoid-related enzymatic components, with 11 DEGs exhibiting distinct stage-specific expression trends (Figure 7). In the early stages of the MVA pathway, with the exception of *HMGS2* and *HMGR1*, other key enzymes exhibited their highest expression levels during the stationary phase (S21) of fungal growth. Between *HMGS1* and *HMGS2*, the expression level of *HMGS1* was higher than the latter, with HMGS1 showing notably higher expression in the stationary phase compared to the exponential phase (S14). Similarly, the expression level of *HMGR2* was markedly higher than that of *HMGR1*, and *HMGR2* demonstrated significantly higher expression in the stationary phase than in the exponential phase. In the process of terpenoid backbone synthesis, *FPPS3* exhibited the highest expression level among all *FPPS* enzymes, followed by *FPPS5* and *FPPS6*. The expression of *FPPS5* was significantly higher during the lag phase (S7) of *A. areolatum* growth compared to the stationary phase, and the expression of FPPS6 during the lag phase was notably higher than in subsequent growth stages (S14 and S21). In triterpenoid synthesis, the expression of *SQLE2* at the stationary phase was significantly higher than at the exponential phase, and *SQLE1* also showed relatively high expression at 21 days. Regarding sesquiterpene synthases, both *AaTPS1* and *AaTPS3* had the highest overall expression level, with significantly higher expression at the stationary phase compared to lag phase and exponential phase. *AaTPS2* reached its maximum expression levels at the stationary phase, which were significantly higher than its expression levels at 7 days. The expression of *AaTPS4* at the stationary phase was significantly lower than at the lag phase and exponential phase, while *AaTPS6* exhibited significantly lower expression at the lag phase compared to subsequent growth stages.

### 3.7. Correlation Analysis of Terpenoids and Differentially Expressed Terpene Synthase Genes

To investigate the association between sesquiterpenoid constituents in *A. areolatum* and transcriptome-annotated sesquiterpene synthetase gene expression, we conducted Pearson correlation analyses between five differentially expressed terpenoid synthetase genes and thirty sesquiterpenoid metabolites using the psych package in R (v4.4.3). The abundance of Sesqui-T1-3 (abscisic acid, eremopetasidione, and dehydrocurdione), Sesqui-T13 (4’-dihydroabscisic acid), and Sesqui-T17 (sterebin B) showed strong positive correlations with the expression levels of *AaTPS2* and *AaTPS6*, with correlation coefficients ranging from 0.82 to 0.95. Sesqui-T4 (13-nor-6-eremophilene-8,11-dione), Sesqui-T8-12 (ketosantalic acid, glandulone C, (4-hydroxy-3-methoxy-2,10-bisaboladien-9-one), (3,11,12-trihydroxy-1(10)-spirovetiven-2-one), and melleolide B), Sesqui-T15 (cryptomeridiol 11-rhamnoside), Sesqui-T19-23 (macrocarpal I, zedoarondiol, glycinoeclepin C, (S)-abscisic acid, and melleolide L), and Sesqui-T30 (2,2,8-trimethyltricyclo[6.2.2.01,6]dodec-5-ene) exhibited significant positive correlations with *AaTPS1-3* (coefficients: 0.89~1.00). Notably, with the exception of Sesqui-T14 (nivalenol), all these sesquiterpenoids displayed highly negative correlations with *AaTPS4* in Figure 8A. In contrast, *AaTPS4* only showed positive correlations with Sesqui-T25 (beta-bisabolol) and Sesqui-T29 (trans-calamenene). Sesqui-T5 (7,9-illudadiene-3,14-diol) and Sesqui-T26 (alpha-curcumene) demonstrated strong positive correlations exclusively with *AaTPS6* (coefficients > 0.97). Sesqui-T6 (5beta-1,3,7(11)-eudesmatrien-8-one), Sesqui-T7 ((7b,10a)-3-hydroxy-1,3,5-cadinatrien-9-one), Sesqui-T16 (4,5-dihydrovomifoliol), and Sesqui-T18 (dolichyl b-D-glucosyl phosphate) exhibited significant positive correlations with the other four differentially expressed genes (DEGs), excluding *AaTPS4*.

### 3.8. Verification of Gene Expression

Transcriptomic validation was performed through the RT-qPCR quantification of six terpenoid biosynthetic genes. Expression profiles demonstrated high concordance between RNA-seq and experimental validation (Figure 9), with the exception of *AaTPS5* temporal expression divergence between days 7 and 14 post-inoculation.

## 4. Discussion

In this study, the integrated application of LC-MS and SPME-GC-MS revealed 103 terpenoids across different developmental stages of *A. areolatum*. This approach significantly expanded the diversity of detected metabolites compared to previous studies relying solely on GC-MS-based methods for analyzing fungal metabolites [3,15,16,17]. Based on the sampling timeline, terpenoid abundance in *A. areolatum* was highest during the stationary growth phase. LC-MS-identified differential metabolites across comparison groups exhibited a consistent upregulation trend in most groups (S14 vs. S7: 28/33, S21 vs. S14: 29/32, and S21 vs. S7: 40/45), likely due to the developmental regulation of terpenoid production. Regarding the higher proportion of downregulated differentially expressed metabolites (DEMs) in GC-MS-detected terpenoids across comparisons (S14 vs. S7: 5/7, S21 vs. S14: 4/5, S21 vs. S7: 7/9), this trend may be attributed to the lower molecular weights of monoterpenoids and sesquiterpenoids, leading to greater volatility and earlier loss. In terms of terpenoid diversity, triterpenes exhibited the highest total abundance throughout the growth cycle of *A. areolatum*, followed by sesquiterpenes, aligning with the terpenoid compositional profile typical of basidiomycetes. Among the identified compounds, antimicrobial agents such as abietinal, melleolide B, and retapamulin were detected, which may contribute to creating a favorable microenvironment for the colonization of *S. noctilio* symbiotic fungi. Additionally, terpenoids, including alpha-curcumene, 2-carene, piperitone, 2,2,8-trimethyltricyclo[6.2.2.01,6]dodec-5-ene, and beta-bisabolol, were identified. These compounds are known to attract *S. noctilio* for oviposition, acting as chemical signals for host selection while simultaneously facilitating fungal infection and dispersal [3,15,16,17]. It should be noted that the terpenoid compounds in this study were analyzed using HRMS. However, the current results should be considered preliminary due to the inherent limitation that isomeric compounds may exhibit identical molecular formulas and similar mass fragmentation patterns. Definitive structural confirmation necessitates complementary approaches, such as nuclear magnetic resonance (NMR) spectroscopy or direct comparison with authentic chemical standards. Future studies need to elucidate their stereochemical configurations through advanced NMR analyses, thereby resolving ambiguities arising from isomeric complexity.

In nature, terpenoids are primarily synthesized through two distinct metabolic pathways: the mevalonate (MVA) pathway and the methylerythritol phosphate (MEP) pathway. However, in basidiomycetes, terpenoid biosynthesis predominantly relies on the MVA pathway [21,37,38]. Following the completion of its genomic sequencing, several terpene synthases (TPS) have been identified in *A. areolatum*. However, a comprehensive understanding of other enzymatic components governing terpenoid biosynthesis in this fungus remains elusive. In this study, we annotated nine key enzymes involved in the terpenoid backbone biosynthesis pathway upstream of terpenoid synthesis, including *AACT*, *HMGS1*, *HMGS2*, *HMGR1*, *HMGR2*, *MVK*, *PMK*, *MVD*, and *IDI*. During the stationary phase of *A. areolatum* growth, the MVA pathway was fully activated, indicating a stronger supply of terpenoid precursors. Notably, *HMGS1* and *HMGR2* exhibited significant upregulation during the stationary growth phase of *A. areolatum*, suggesting their pivotal roles in driving terpenoid accumulation at this developmental stage. This observation aligns with previous findings in *Rhodosporidiobolus colostri*, where upregulation of the terpenoid backbone pathway enhanced the production of β-carotene and torulene [35]. Furthermore, a k-means clustering analysis of all DEGs demonstrated that 116 DEGs in Subclass 3 exhibited similar expression patterns with *HMGS1* and *HMGR2* during the growth phase of *A. areolatum*, suggesting potential synergistic interactions with these key enzymes in terpenoid biosynthesis.

Furthermore, through KEGG database annotation, we identified seven farnesyl pyrophosphate synthases (*FPPS*) and one geranylgeranyl pyrophosphate synthase (*GGPPS*). These enzymes mediate the condensation of isopentenyl pyrophosphate (IPP) and dimethylallyl pyrophosphate (DMAPP) into higher-order isoprenoid diphosphates (e.g., farnesyl pyrophosphate [FPP], geranylgeranyl pyrophosphate [GGPP]), which act as direct precursors for diverse terpenoid classes, including sesquiterpenes, diterpenoids, and triterpenes. Notably, *FPPS* expression was significantly expressed in the 7-day post-inoculation samples, ensuring a robust supply of FPP precursors to support terpenoid biosynthesis. Regarding triterpenoid metabolite biosynthesis, we annotated one farnesyl-diphosphate farnesyltransferase (*FDFT*) and two squalene monooxygenases (*SQLE*), with *SQLE* genes exhibiting the highest expression levels in 21-day post-inoculation samples. This upregulated expression of *SQLE* likely serves as a key driver for the substantial accumulation of triterpenoids during the stationary growth phase of *A. areolatum*.

Terpene synthases (TPS) are abundant in basidiomycetes. For example, *Coprinus cinereus* encodes six *STS* [39], *Omphalotus olearius* encodes eleven *STS* [40], and *Agrocybe aegerita* encodes nine *STS* [41]. As demonstrated in Figure 3, the NR database exhibited a significantly higher annotation capacity (92.12%) compared to the KEGG database (37.19%) during the functional annotation of *A. areolatum* transcriptomic data. To address the limitations of the KEGG database in annotating fungal terpenoid biosynthesis genes, complementary annotation of terpene synthase (TPS) genes was performed using the NR database based on homology. *A*. *areolatum* encodes six STS in this study, suggesting that *A. areolatum* exhibits a typical level of TPS production. Among the six sesquiterpene synthase (*STS*) genes analyzed, five were identified as differentially expressed genes (DEGs), with the transcript levels of *AaTPS1-3* exhibiting a progressive increase during fungal cultivation. Correlation analysis between the five DEGs and sesquiterpenoid metabolites revealed distinct associations: *AaTPS4* displayed a strong positive correlation with beta-bisabolol and trans-calamenene, while *AaTPS6* showed high positive correlations with 11 sesquiterpenoids. Notably, *AaTPS1-3* demonstrated significant positive correlations with 17 sesquiterpenoids, suggesting their pivotal regulatory roles in sesquiterpenoid biosynthesis. Additionally, 467 DEGs exhibited congruent expression profiles with *AaTPS1-3* in k-means clustering analysis of DEGs, potentially facilitating the regulation of sesquiterpenoid biosynthesis pathways through coordinated transcriptional control. To investigate the biosynthesis of beta-bisabolol, alpha-curcumene, and 2,2,8-trimethyltricyclo[6.2.2.01,6]dodec-5-ene in *A. areolatum*, targeted manipulation of DEGs with high correlation coefficients could be implemented. For instance, *AaTPS4* (strongly positively correlated with beta-bisabolol) and *AaTPS1-3* (negatively correlated with beta-bisabolol) could be modulated to dissect their opposing regulatory effects. Similarly, *AaTPS6* (highly positively correlated with alpha-curcumene) and *AaTPS1-3* (positively correlated with 2,2,8-trimethyltricyclo[6.2.2.01,6]dodec-5-ene) may serve as key candidates for functional validation, whereas *AaTPS4* (negatively correlated with 2,2,8-trimethyltricyclo[6.2.2.01,6]dodec-5-ene) might act as a suppressor. In research related to fungal terpenoid biosynthesis, regulating terpenoid production through the MVA pathway, gene expression modulation, and cofactor optimization remain a focus of current studies [42]. In future work, functional validation and gene knockout of these terpenoid biosynthesis-related genes are necessary to establish a theoretical foundation for the practical production of terpenoid compounds in *A. areolatum*.

## 5. Conclusions

In this study, 103 terpenoid compounds were detected in *A. areolatum* using LC-MS and SPME-GC-MS, significantly enriching the metabolic dataset of *S. noctilio* symbiotic fungi. The total terpenoid abundance progressively increased during fungal growth, spanning the lag, exponential, and stationary phases. Through functional classification and gene annotation, we identified 26 genes associated with terpenoid biosynthesis in the mevalonate (MVA) pathway of *A. areolatum*, including 11 differentially expressed genes (DEGs) (|log2Fold Change| ≥ 1, adjusted *p*-value < 0.05). Notably, *HMGS1* (*Gene_EVM002515*) and *HMGR2* (*Gene_EVM010955*) likely play pivotal roles in terpenoid backbone formation, contributing to the accumulation of fungal terpenoids during the stationary phase. Furthermore, *AaTPS1-3* (*Gene_EVM006403*, *Gene_EVM006404*, and *Gene_EVM002513*) were found to exert particularly crucial regulatory effects on sesquiterpenoid synthesis. Future studies should focus on the functional validation of these candidate genes to confirm their roles in terpenoid biosynthesis. Additionally, investigating the synergistic effects of environmental factors (e.g., nutrient availability or host interactions) on the MVA pathway could further elucidate the ecological relevance of terpenoid production in *A. areolatum*.

## Figures and Tables

**Figure 1 jof-11-00383-f001:**
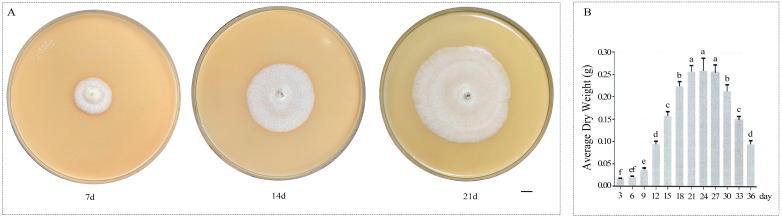
Growth dynamics of *Amylostereum areolatum*. (**A**) Colony morphology of *A. areolatum* cultured on 15 cm Petri dishes for 7, 14, and 21 days. Scale bar = 1 cm. (**B**) Growth rate profile of *A. areolatum* based on biomass dry weight determination [33]. Error bars represent the standard deviation (SD), and different lowercase letters indicate a significant difference among the average dry weight of treatments (*p* < 0.05).

**Figure 2 jof-11-00383-f002:**
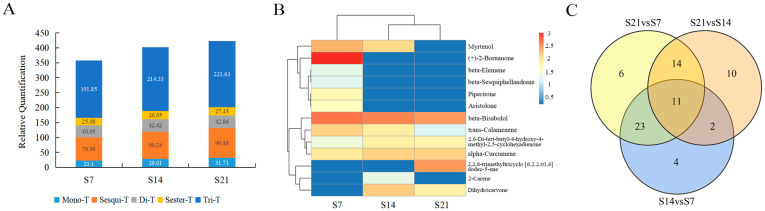
Distribution of terpenoid abundance across lag phase, exponential phase, and stationary phase, along with metabolic variations among these growth stages. (**A**) Total abundance of different terpenoids of *A. areolatum* detected using LC-MS at 3 stages. (**B**) Sample cluster map of terpenoids of *A. areolatum* detected using GC-MS. Sample identifiers and metabolite clusters are plotted on the *x*-axis and *y*-axis, respectively, with color gradients indicating the values obtained after the relative content standardization process. (**C**) The number of significantly different terpenoids among groups. S7, S14, and S21: Sample of *A. areolatum* grown in damaged Mongolian Scots pine wood powder medium during lag phase, exponential phase, and stationary phase, respectively.

**Figure 3 jof-11-00383-f003:**
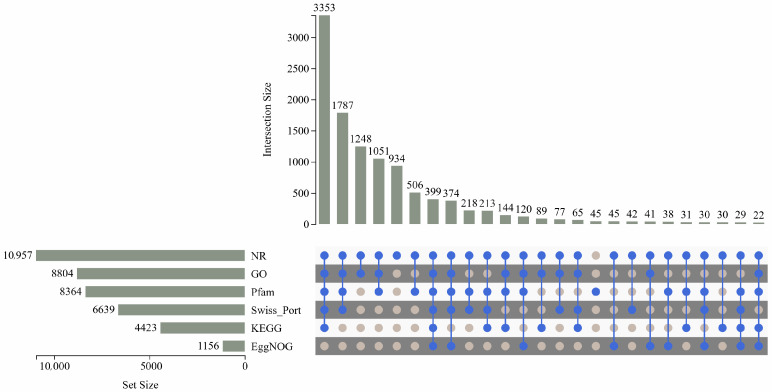
Annotation information for all expressed genes obtained from the transcriptome assembly of *A. areolatum*. The bar graph at the bottom left represents the distribution of annotation information from different public databases. Different filling methods in the lower right corner represent the intersection of annotation information in different databases, and the upper right corner represents the amount of data that all databases jointly annotate for genes.

**Figure 4 jof-11-00383-f004:**
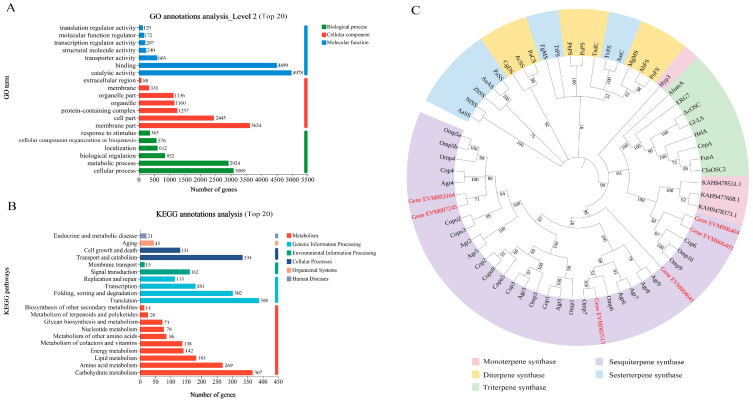
Annotation information for RNA-Seq. (**A**) GO annotation classification map of *A. areolatum* transcriptome data (Top 20). The ordinate corresponds to functional subcategories within the GO, while the abscissa displays gene count distribution, and the colors indicate different ontology categories. (**B**) KEGG pathway classification map of *A. areolatum* transcriptome data (Top 20). The ordinate is the name of the KEGG metabolic pathway. The abscissa is the number of genes annotated to this pathway, and the colors indicate different categories in KEGG. (**C**) Phylogeny tree analysis of *A. areolatum* TPSs. The phylogenetic tree was generated in MEGA-12 with 1000 bootstrap replicates using the Neighbor-Joining algorithm. The protein sequences used in the phylogenetic tree were obtained from NCBI and JGI, and the accession number is listed in Appendix A.

**Figure 5 jof-11-00383-f005:**
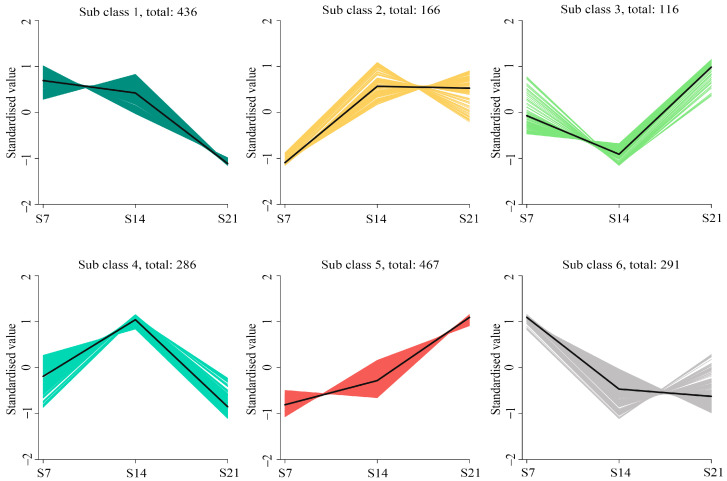
Clusters of differentially expressed genes (DEGs). k-means clustering partitioned the DEGs into six expression subclasses (Subclass 1–Subclass 6). Sample identifiers and standardized expressions are mapped to the abscissa and ordinate, respectively.

**Figure 6 jof-11-00383-f006:**
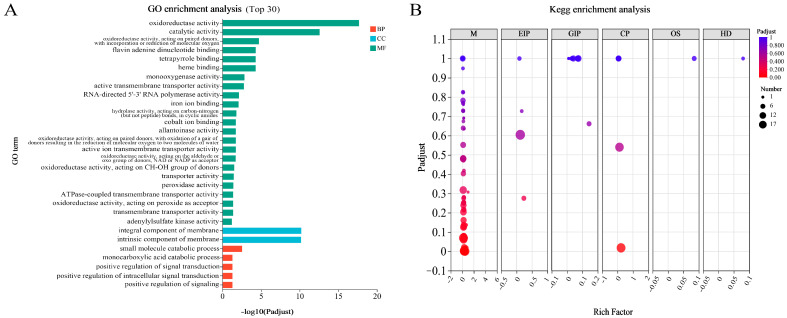
Enrichment map of DEGs. (**A**) GO enrichment histogram of DEGs. The vertical coordinate represents the GO term, and the horizontal coordinate represents the significance level of enrichment. All DEGs were divided into three GO entries: biological process (BP), cellular component (CC), and molecular function (MF). (**B**) KEGG enrichment analyses of DEGs. The horizontal coordinate represents the enrichment rate, and the vertical coordinate represents the significance level of enrichment. M: metabolism, EIP: environmental information processing, GIP: genetic information processing, CP: cellular processes, OS: organismal systems, HD: human diseases.

**Figure 7 jof-11-00383-f007:**
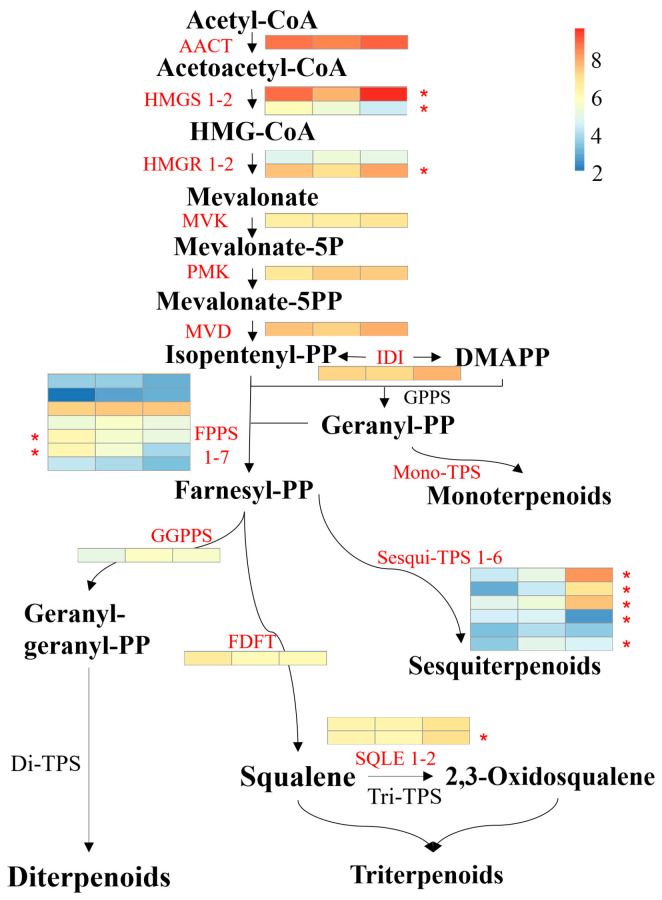
Expression patterns of terpenoid biosynthesis genes. The expression pattern of each gene is displayed in a three-column grid; 7-, 14-, and 21-day samples are presented from left to right. The color scale represents the log2-fold change in gene expression levels. AACT, acetyl-CoA acetyltransferase; HMGS, hydroxymethylglutaryl-CoA synthase; HMGR, hydroxymethylglutaryl-CoA reductase; MVK, mevalonate kinase; PMK, phosphomevalonate kinase; MVD, diphosphomevalonate decarboxy lase; IDI, isopentenyl-diphosphate delta-isomerase; DMAPP, dimethylallyl diphosphate; GPPS, geranyl diphosphate synthase; FPPS, farnesyl diphosphate synthase; GGPPS, geranylgeranyl diphosphate synthase; FDFT: farnesyl diphosphate farnesyltransferase; SQLE: squalene monooxygenase; Mono-TPS: monoterpene synthase; Sesqui-TPS: sesquiterpene synthase; Di-TPS: diterpene synthase; Tri-TPS: triterpene synthase. * represents DEGs.

**Figure 8 jof-11-00383-f008:**
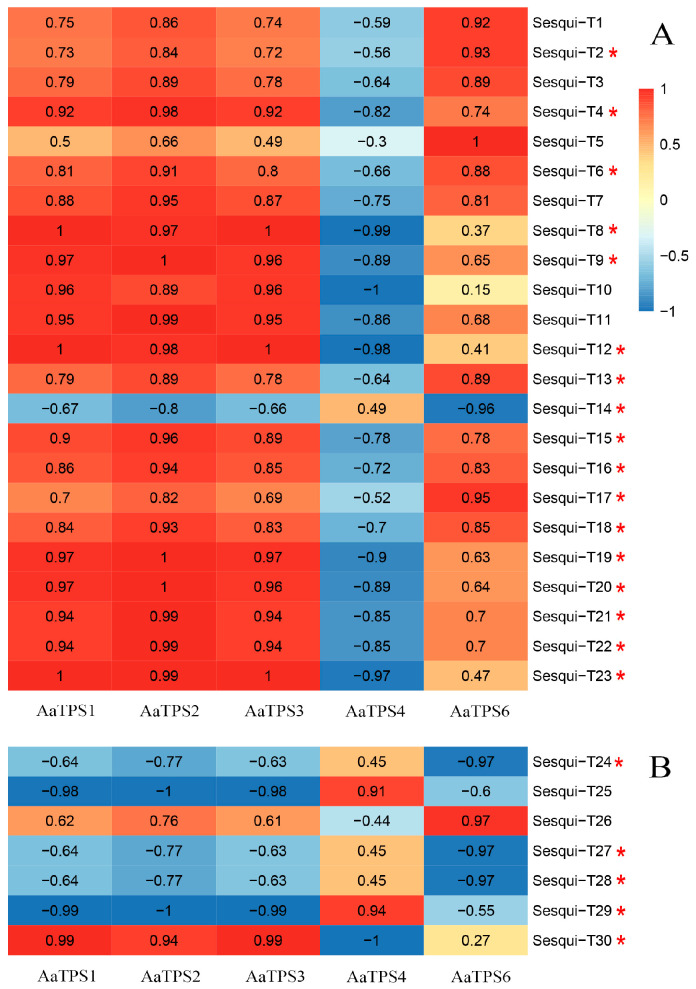
Correlation analysis of sesquiterpenoids and differentially expressed sesquiterpene synthase genes. (**A**) Correlation between differentially expressed sesquiterpene synthase genes and sesquiterpenoids detected using LC-MS; (**B**) Correlation between differentially expressed sesquiterpene synthase genes and sesquiterpenoids detected by GC-MS. * represents significantly different produced terpenoids. The number and corresponding color scale illustrate the relationships between sesquiterpenoid and sesquiterpene synthetase genes.

**Figure 9 jof-11-00383-f009:**
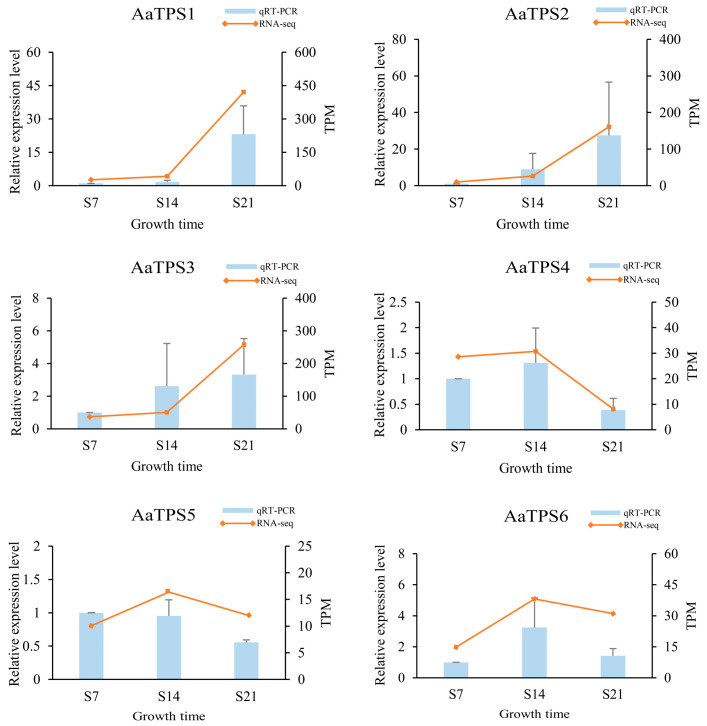
Expression patterns verified via RT-qPCR of terpenoid synthesis genes at different developmental stages. The RT-qPCR results (columns) are compared with RNA-Seq data (lines). Standard deviations of the means for three biological replicates are represented by the error bars.

**Table 1 jof-11-00383-t001:** Quality statistics of transcriptome sequencing data.

Sample ID	Clean Reads	Clean Bases (bp)	Error Rate (%)	GC Content (%)	Q20 (%)	Q30 (%)	Mapping Ratio (%)
S7_1	51,889,968	7,670,121,267	0.0253	57.46	97.82	93.98	89.37
S7_2	43,616,060	6,412,761,653	0.0251	57.54	97.89	94.19	87.81
S7_3	48,759,422	7,206,339,669	0.0252	57.23	97.85	94.02	88.63
S14_1	46,067,288	6,825,290,603	0.0251	57.19	97.88	94.11	89.01
S14_2	40,644,990	6,012,925,517	0.0251	57.4	97.88	94.16	89.61
S14_3	46,154,948	6,849,932,991	0.0253	57.27	97.81	93.98	89.21
S21_1	47,690,676	7,005,804,935	0.0249	57.45	97.98	94.38	91.95
S21_2	47,365,588	6,963,548,962	0.0248	57.28	97.99	94.41	91.6
S21_3	45,241,166	6,714,329,801	0.0251	57.31	97.89	94.19	90.66

**Table 2 jof-11-00383-t002:** Statistics of DEGs during terpenoid synthesis.

No.	Comparison	Gene ID	Name	Fold Change	Log2FC	*p* Value	*p*-Adjust	Type
1	S14 vs. S7	*Gene_EVM002515*	*HMGS1*	0.4172	−1.2610	0.0000	0.0001	down
2		*Gene_EVM007970*	*FPPS6*	0.4899	−1.0294	0.0034	0.0405	down
3		*Gene_EVM006404*	*AaTPS2*	2.2842	1.1917	0.0016	0.0244	up
4		*Gene_EVM007245*	*AaTPS6*	2.2338	1.1595	0.0000	0.0001	up
5	S21 vs. S14	*Gene_EVM002515*	*HMGS1*	4.4473	2.1529	0.0000	0.0000	up
6		*Gene_EVM010955*	*HMGR2*	2.3009	1.2022	0.0033	0.0312	up
7		*Gene_EVM007970*	*FPPS6*	0.3928	−1.3481	0.0012	0.0152	down
8		*Gene_EVM008329*	*SQLE2*	2.3181	1.2130	0.0001	0.0025	up
9		*Gene_EVM006403*	*AaTPS1*	11.2815	3.4959	0.0000	0.0000	up
10		*Gene_EVM002513*	*AaTPS3*	5.8442	2.5470	0.0052	0.0435	up
11		*Gene_EVM003164*	*AaTPS4*	0.2998	−1.7378	0.0000	0.0000	down
12	S21 vs. S7	*Gene_EVM002681*	*HMGS2*	0.3376	−1.5668	0.0000	0.0003	down
13		*Gene_EVM007969*	*FPPS5*	0.4732	−1.0793	0.0030	0.0182	down
14		*Gene_EVM007970*	*FPPS6*	0.1928	−2.3752	0.0000	0.0000	down
15		*Gene_EVM006403*	*AaTPS1*	15.7895	3.9809	0.0000	0.0000	up
16		*Gene_EVM006404*	*AaTPS2*	16.7023	4.0620	0.0001	0.0009	up
17		*Gene_EVM002513*	*AaTPS3*	6.9804	2.8033	0.0013	0.0090	up
18		*Gene_EVM003164*	*AaTPS4*	0.2760	−1.8573	0.0000	0.0000	down
19		*Gene_EVM007245*	*AaTPS6*	2.0777	1.0550	0.0001	0.0011	up

## Data Availability

All data mentioned in this paper are available at the National Center for Biotechnology Information (NCBI) under BioProject no. PRJNA1162567. Raw sequence data (Illumina, resequencing and RNA-seq) have been deposited in the NCBI Sequence Read Archive (SRA) database under BioSample no. SRR30728529-SRR30728537. RNA-seq sequencing data of *Amylostereum areolatum* at different growth stages have been transmitted under BioSample no. SAMN43816638-SAMN43816646.

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
