# Peer review of "Integrative Transcriptome and Metabolome Analysis Reveals Candidate Genes Related to Terpenoid Synthesis in Amylostereum areolatum (Russulales: Amylostereaceae)"

_jof, 2025, doi:10.3390/jof11050383_

Round 1

Reviewer 1 Report

See attached.

See attached.

Author Response

Response to Reviewer 1 Comments

Dear Reviewer,

Thank you for your comments on our manuscript. Those comments are very helpful for revising and improving our paper, as well as the important guiding significance to our research. We have studied the comments carefully and made corrections which we hope meet with approval. All corrections are marked in red in the manuscript. The responses to the reviewers’ comments are as follows (the replies are highlighted in red).

1. Metabolite identification was carried out via LC-HRMS (but not MS/MS), and GC-MS, and then searching databases such as HDMB and Metlin for matches. I have several concerns with this approach, which I will list below:

a. First and foremost, terpenoids are notoriously difficult to identify because compounds with the exact same chemical formula and m/z ratio can have many different chemical structures. While grouping them into subclasses (e.g., monoterpenoids, diterpenoids, etc.) is relatively straightforward, HRMS is not enough for unambiguous identification of terpenoids. Therefore, the authors should at least acknowledge this in the main text and highlight that what they find are putative The only way to confirm the compounds’ identity would be to isolate them and carry out NMR analyses.

Response: Thank you for your valuable comments. We have acknowledged the limitations in terpenoid compound identification within the Discussion section, as detailed in lines 513-520 of the revised manuscript.

b. Why were natural products databases, such as NP Atlas, Coconut, or GNPS, not included in the list of searched databases? These are often used for microbial natural products and might have led to identification of additional

Response: We are very sorry for our carelessness. In this study, we have conducted GNPS database searches as part of our analytical approach. The relevant methodological details have been incorporated into the revised version (lines 167-169). In addition, NP Atlas and Coconut were utilized as conceptual references; they neither provide directly accessible reference spectra. Specifically, NP Atlas requires GNPS database support for spectral matching. Therefore, our LC-MS based metabolite identification was performed using a comprehensive approach combining HMDB, Metlin, GNPS databases, and the proprietary database maintained by our contracted metabolomics service provider.

c. The authors report the identification of 792 fungal metabolites from LC-MS, which is a rather large number. How was the data processed? Were peaks below a certain threshold excluded? Were all the peaks that appeared also in the blank samples excluded as well. Please clarify this and add these details in the experimental 

Response: Thank you for your valuable comments. The LC-MS derived data underwent systematic preprocessing, where the resultant data matrix was aligned with metabolite databases for putative identification. Metabolite features with signal-to-noise ratio (S/N) ≥ 3 were retained as qualified annotations. Background interference was rigorously controlled through sum normalization of blank samples, coupled with peak filtering and QC validation. We have substantially expanded the methodological description of this workflow in Section 2.2 (lines 162-176).

d. Full chromatograms (TIC and/or BPI) of the fungal extracts (possibly also compared to blank medium) should be included in the supplementary information, at least of one representative sample per group. This also helps reviewers and readers to see if the extracts are indeed so complex as they seem from the sheer number of metabolites identified, or if there might be flaws in the analysis 

Response: Thank you for this constructive suggestion. By your recommendation, we have now included the Total Ion Chromatograms (TIC) of Amylostereum areolatum samples both GC-MS and LC-MS analyses as Supplementary Figures S1 and S2, respectively. Thank you.

2. In my opinion, most figures are not of publication quality at this stage:

a. The font size in both main and supplementary figures is very small, which makes it difficult to read without zooming in. I suggest that the authors increase the font size to match at least the minimum required by the journal guidelines.

Response: Thank you for your valuable comments. In the revised manuscript, we have enlarged the font sizes in Figure 1B, Figure 4, Figure 6, Figure S4 (previously Figure S2), and Figure S5 (originally Figure S3) to enhance readability.

b. Figure 1 B is severely skewed and not sharp. Also, it almost looks as though it was photographed or snapped from another manuscript. If that is the case, a reference and permission from the authors should be

Response: Thank you for your comments. We have revised Figure 1B to improve its clarity. The Figure 1B was generated by our research group, and we have obtained proper authorization from both the supervisor and the author.

c. Figure7 is also very unsharp, and a bit confusing in some cases: for example, the STS matrix is now placed right below the label “monoterpenoids” which might lead to misinterpretation. Also, it is not mentioned what the colour scale represents, log2 fold change, fold change? Please, clarify and adjust the figure for better 

Response: Thank you for your suggestion. The clarity of Figure 7 has been improved. We have repositioned the STS matrix in the revised Figure 7. Additionally, the color scale legend has been incorporated into the figure caption with the annotation: 'The color scale represents the log2-fold change in gene expression levels’(lines 416-417).

3. It is not clear whether the samples for transcriptomics and metabolomics were taken from liquid or solid cultures. This is an important detail to include, because for liquid cultures one can assume that sampling biomass at a given timepoint does indeed give a picture of that growth phase. Whereas for cultures on agar, this is entirely different, and it depends on where the sampling as been done: for example, at the outer edge the fungus is likely in exponential phase, while towards the centre of the colony, the fungus will be in stationary phase or even metabolically inactive. Please clarify this aspect, and, if cultures grown on solid media were used, describe the rationale behind the sampling and how it relates to the results.

Response: We appreciate the reviewer's comments. Current research data indicate that the formation of fruiting bodies by A. areolatum has not been documented in natural habitats within China. And as detailed in Section 2.1 (lines 114-118), our experimental design intentionally mimicked the natural habitat of A. areolatum. The solid culture medium was prepared using pine wood powder collected from the edge of Sirex noctilio burrows in infested Pinus sylvestris var. mongolica. Fungal cultivation and sampling were conducted on this substrate to better approximate natural ecological conditions, thereby enhancing the environmental relevance of our findings.

4. Section 3.4 is very brief. It would be helpful if the authors expanded the text to clarify what these results entail. Perhaps explain in a sentence or two what k-means clustering means as not every reader might be familiar with this type of

Response: Thank you for your valuable comments. We have incorporated detailed explanations of the k-means clustering analyses for DEGs in both Section 3.4 (Results) and the Discussion section. Including (1) The expression pattern of specific DEGs clusters (lines 372-377), (2) the prediction of the synergistic effect between specific DEGs clusters and important terpene synthesis-related genes (lines 536-539, and lines 570-573).

5. Insection 7, the authors state “we conducted correlation analyses between five differentially expressed terpenoid synthetase genes and thirty sesquiterpenoid metabolites” and then proceed to discuss the results. It would be helpful to re-discuss briefly how this analysis was done and what the numbers mean. This will facilitate interpretation of data and results.

Response: Thank you for your comments. We have added explanations for the numerical labels in the figure caption of Figure 8 and incorporated the Pearson correlation analysis methodology in Section 3.7. Please refer to lines 456-457 and 460-461 for details.

6. In table S1 and S2 several numbers are shown to indicate abundance/amount of each terpenoid I assume, but there is no information as to what those numbers  Are they based on signal intensity (total ion or extracted ion count), or peak area, or something else? Please clarify and add this information.

Response: Thank you for your suggestion. We have added information on the content of terpenoid compounds to the notes of Supplementary Tables S1 and S2.

Table S1: Note: The abbreviations stand for: Mono-T = Monoterpenoids, Sesqui-T = Sesquiterpenoids, Di-T = Diterpenoids, Sester-T = Sesterterpenoids, Tri-T = Triterpenoids. The relative abundances of terpenoid metabolites quantified by LC-MS in the table were derived from raw data subjected to preprocessing (baseline correction, peak alignment, and retention time calibration) followed by total sum normalization. Normalized values were log10-transformed to obtain the dataset. In contrast, terpenoid abundances measured by GC-MS were obtained through internal standard normalization using 2-Octanol,(S)- (50ug/mL) as the reference compound, with raw data processed via 7890B-5977B GC-MS system.

Table S2: Note: The relative abundances of terpenoid metabolites quantified by LC-MS in the table were derived from raw LC-MS data subjected to metabolite annotation (HMDB, Metlin, and GNPS libraries) and peak area extraction. Missing values were imputed with the minimum positive value detected across the entire dataset. In contrast, terpenoid abundances measured by GC-MS were obtained through internal standard normalization using 2-Octanol,(S)- (50ug/mL) as the reference compound, with raw data processed via 7890B-5977B GC-MS system.

7. Lines 295-296: it is not clear how the authors determined the number of total genes. Transcriptomics measures the levels of transcript RNAs in the cell and therefore identifies only the expressed genes. How did the authors obtain the total number of genes and discriminate between unexpressed and expressed genes? Please clarify and expand the sentences introducing this section.

Response: Thank you for your suggestion. As this study is a reference genome-based transcriptome study, the total gene count was derived exclusively from known genes in the reference genome. Expressed genes refer to those with detectable expression in at least one sample, while non-expressed genes are defined as having zero expression across all samples. This clarification has been added to the main text (see lines 313-316).

8. I suggest italicizing scientific names up to division rank, as suggested by the International Code of Nomenclature for algae, fungi, and plants (chapter III). https://www.iapt-taxon.org/nomen/main.php.

Response: Thank you for your suggestion. We have italicized the scientific names of fungi in the manuscript. Please refer to line 47 for the updated formatting.

9. According to the Westerdijk Institute (h  ttps://wi.knaw.nl/) the fungus is now classified in a different family, see below. Given that WI is an authority in fungal taxonomy and provides the most up-to-date information, I would suggest modifying accordingly.

Amylostereum areolatum (Chaillet ex Fr.) Boidin, Revue de Mycologie (Paris) 23 (3): 345 (1958) [MB#292521] – Fungi Dikarya Basidiomycota - Agaricomycotina - Agaricomycetes - Russulales - Amylostereaceae - Amylostereum.

Response: We sincerely apologize for the oversight. A. areolatum indeed belongs to the order Russulales and the family Amylostereaceae. We have corrected this in the manuscript. Please refer to the revised title, Line 15, and Lines 46-47 for the updated taxonomic classification.

10. There are sometimes spacing typos before or after in text references, please adjust (e.g., line 41, line 86, lines 93 – 94).

Response: We are very sorry for our carelessness. We have corrected the errors you identified and conducted a comprehensive review of the entire manuscript to ensure accuracy.

11.Lines 45-46: the taxonomy here is different, please correct.

Response: We are very sorry for our carelessness. A. areolatum belongs to the family Amylostereaceae, and we have corrected this in the manuscript. Please refer to lines 46-47 for the updated information.

12. Some typos: line 48, symbiotic fungi are stored (or localize); line 63, conducted volatile profile s tudies; line 87, synthaseS plays; lines 411 and 412, their  →  its; line 232, Results 

Response: We sincerely apologize for the oversight. We have corrected these errors in the revised manuscript. The corresponding updates can be found in Line 49, Line 65, Line 89, Lines 445-446, and Line 246 for your review.

13. Line 146: i.d. of the column is given in µM(micromolar) instead of µm (micrometer), please correct.

Response: Thank you for your comments. We have reviewed the manuscript and added ‘(micromolar)’ after ‘µM’ as supplementary clarification. Please refer to Line 151 in the revised manuscript.

14. Lines 155-156: please clarify if the voltages reported refer to capillary

Response: We sincerely apologize for the oversight. The term "voltages" here specifically refers to spray voltage, and we have clarified this in the revised manuscript. Please see Line 160 for the updated context. Thank you for your attention to detail.

15. Some text links are in a different font and underlined in blue, e.g., line 165, line 224, and many links in the references. Please fix

Response: Thank you for your comments. We have standardized the formatting of all hyperlinks throughout the manuscript, ensuring consistent font type and color scheme, and removing underlines. This revision has been uniformly applied across the manuscript.

16. Report the LC method as it was done for GC method. That is, with the wording “increase to, heldat, etc”. This is a more standard why to describe chromatography methods and more easily 

Response: Thank you for your comments. We have revised the phrasing in Line 152 by replacing "maintained at" with "held at" to better align with the technical context.

17. Lines 188-189: significant differences terpenoids → do you mean significantly differently produced? Also, in lines 421-422 there is a similar wording, please 

Response: Thank you for your comments. We have revised the phrasing to "significantly differently produced terpenoids" to better reflect the experimental observations. This adjustment has been incorporated in Line 200 and Line 456 of the revised manuscript.

18. Line199: What do the authors mean with “To establish analytical rigor”?  In general, the sentence needs to be adjusted as it does not read smoothly.

Response: Thank you for your comments. We have removed the phrase "To establish analytical rigor" and revised the sentence as follows:

First, the software SeqPrep (https://github.com/jstjohn/SeqPrep) and Sickle (https://github.com/najoshi/sickle) were used for raw data quality control, and then HISSAT2 was used to compare the raw clean data (reads) with the A. areolatum reference genome (INSDC: SAXG00000000.1; BioProject: PRJNA513942) to obtain mapped data (reads) [30].

19. Lines 209-213: each database should be referenced properly according to the journal’s guidelines

Response: Thank you for your comments. We have added the relevant database URLs to the manuscript as recommended. The updates can be found in Lines 222-227 of the revised version.

20. Line 218: is the 0.8 threshold arbitrary or chosen based on previous publications? Please clarify

Response: Thank you for your comments. The threshold of 0.8 was selected based on the published literature. We have now incorporated supporting citations in Section 2.4 of the revised manuscript. The relevant references are as follows:

Xu, X.; Li, C.; Wu, F.;Zhao, S.; Chen, T.; You, H.; Lin, Y.;Zou, X. Integrated Transcriptomic and Targeted Metabolomic Analysis Reveals the Key Genes Involved in Triterpenoid Biosynthesis of Ganoderma lucidum. J. Fungi 2025, 11, 57. https://doi.org/10.3390/jof11010057

Yu, S.; Yang, F.; Pu, J.; Guo, Q.; Zou, Q.; Zhang, H.; Liu, M.; Zheng, Y.; Wang, T. Integrated transcriptomic and metabolomic analyses reveal the characteristics of volatile oils in Chrysanthemum morifolium. Sci Hortic 2023, 315,111983. https://doi.org/10.1016/j.scienta.2023.111983

Li, C.; Xu, Y.; Li, Z.; Cheng, P.; Yu, G. Transcriptomic and metabolomic analysis reveals the potential mechanisms underlying the improvement of β-carotene and torulene production in Rhodosporidiobolus colostri under low temperature treatment. Food Res Int 2022, 156, 111158, doi.org/10.1016/j.foodres.2022.111158.

21. Line 226: why was a p450 used as reference gene for quantitative PCR, and which P450? Please clarify and elaborate this

Response: Thank you for your feedback. Our team member Ningning Fu evaluated the stability of 10 candidate reference genes (α-TUB, β-TUB, γ-TUB, GAPDH, CYP, CPR [NADPH-dependent cytochrome P450 oxidoreductase], HH3 [Histone H3], CESP [coatomer epsilon subunit-domain-containing protein], MDP [metallo-dependent phosphatase], and P450 [cytochrome P450]) across A. areolatum samples at various developmental stages using methods including geNorm, NormFinder, BestKeeper, delta Ct, and RefFinder. Results demonstrated that P450 (cytochrome P450), CYP, and γ-TUB, either individually or in combination, exhibited reliable stability under our experimental conditions. P450 (cytochrome P450) was identified as the most stable gene and was therefore selected as the reference gene for this study. We have now explicitly included P450 (cytochrome P450) in the text with supporting citations. Please refer to Line 240 for the updated content.

22. Line 234: numbers should be spelled out at the beginning of a sentence.

Response: Thank you for your comments. We have added a sentence at the beginning of this paragraph and corrected the spelling of "762" in the revised text. Please refer to Lines 248-250 for the specific revisions.

23. Lines 273-274: here the authors report ratios as fractions, whereas in lines 271-272 they are given as percentages.

Response: Thank you for your comments. We have harmonized to use fractions. The specific revisions can be found in Lines 288-289 and Line 498 of the updated text.

24. Line 285: the wording “in S. noctilio symbiotic fungus” may be confusing, please use the fungus’ name as done across the

Response: Thank you for your comments. We have revised the phrasing by replacing ‘in S. noctilio symbiotic fungus’ with ‘A. areolatum’ throughout the manuscript.

25. In Table 1, thousands separators should be added to the numbers of reads and bp counts to facilitate readability.

Response: Thank you for your comments. We have added thousands of separators to the numbers of reads and bp counts in Table 1.

26. In lines 296-297, the word “annotated” is repeated. Please rephrase.

Response: Thank you for your comments. We have revised the phrasing as follows:

Six databases (NR, PFAM, GO, KEGG, EggNOG, and Swiss-Prot) provided functional annotations for 11,002 genes.

27. Line 325-326: “obtained by annotated in other databases and Blast-P in NR database”. This sentence is unclear. Also, what other database was used? Please clarify and adjust the sentence.

Response: Thank you for your suggestion. We have clarified the annotations related to the databases and revised the corresponding sentences for improved accuracy. Please refer to Lines 345-349 in the updated manuscript for the specific modifications.

28. Line 347: this is a bar chart, not a histogram. Please rephrase.

Response: We sincerely apologize for the unclear phrasing in the original manuscript. Following another reviewer’s suggestion, we have replaced the bar chart with a textual expression. The revised explanation is now provided in Lines 366-370 of the manuscript. Thank you for highlighting this issue.

29. Lines 398-399: “Between HMGS1 and HMGS2, the expression level of HMGS1 was higher than later…”. Do you mean higher than the latter? Please clarify.

Response: We sincerely apologize for the lack of clarity in the original phrasing. The sentence has been revised to: ’The expression level of HMGS1 was higher than the latter.’

30. Line 436: sesqui-T15 should be sesqui-T14 if I understand the table correctly. Please adjust.

Response: We sincerely apologize for the oversight. In the revised manuscript, we have corrected the compound designation by replacing ‘Sesqui-T15 (Cryptomeridiol 11-rhamnoside)’ with ‘Sesqui-T14 (Nivalenol)’ to ensure accuracy. Please refer to Line 432 for the updated content. Thank you for your meticulous review.

31. Figure 9. Check font color in AaTPS3 graph, the font in the legend seems to be grey instead of black. Also, please add information about number of replicates and what the error bars represent in the caption.

Response: We sincerely apologize for the oversight. We have updated the legend color of the AaTPS3 graph to black. Furthermore, the caption of Figure 9 has been revised to include the clarification:

"Standard deviation of the mean for three biological replicates is represented by the error bars."

Reviewer 2 Report

The paper entitled "Integrative transcriptome and metabolome analysis reveals 2 candidate genes...." by Lixia Wang et al. is well done and the results presented by the authors are relevant in this specific area, integrating metabolomic and transcriptomic data to differentially expressed genes in the different fungal growth phases responsible for terpenoids biosynthesis. The manuscript is clearly written and the results are appropriately presented, except for some issues mentioned in the "Detailed comments".

Figure 1: What do the letters in Figure 1B mean? The parts of Figure 1, A and B, are not mentioned in the text.

Line 341: change dientified by identified

Line 342: "Figure 5A shows the distribution of up- and down-regulated DEGs". It is not a distribution (better in a box plot).

Line 344. “All DEGs underwent k-means clustering analysis, categorizing 1,762 genes into six distinct expression clusters (116-467 genes per cluster; Figure 5B)”. Did this analysis have any significance beyond simple categorization? This categorization is not discussed further in the manuscript. The data shown in Figure 5A can be mentioned in the text without the need for a figure. If Figure 5B is not discussed, it is not necessary. Therefore, the entire Figure 5 is not necessary.

In the Discussion section, the first paragraph (lines 454 to 461) is unnecessary. It is mentioned in the Introduction.

Lines 97, 528, 531, 534: change bata to beta

Author Response

Dear Reviewer,

Thank you for your comments on our manuscript. Those comments are very helpful for revising and improving our paper, as well as the important guiding significance to our research. We have studied the comments carefully and made corrections which we hope meet with approval. All corrections are marked in red in the manuscript. The responds to the reviewers’ comments are as follows (the replies are highlighted in red).

Figure 1: What do the letters in Figure 1B mean? The parts of Figure 1, A and B, are not mentioned in the text.

Response: We sincerely apologize for the oversight. We have updated the figure caption of Figure 1 to clarify the meaning of the labels in Figure 1B. Additionally, the main text now explicitly references Figure 1A and Figure 1B separately. These revisions can be found in Line 120 and Line 122 of the revised manuscript. Thank you for your attention to detail.

Line 341: change dientified by identified.

Response: We are very sorry for our carelessness. We have changed dientified by identified.

Line 342: "Figure 5A shows the distribution of up- and down-regulated DEGs". It is not a distribution (better in a box plot).

Response: Thank you for your comments. In accordance with your subsequent comment, we have removed the bar chart and instead described the expression profile classification of DEGs textually within the manuscript. Please refer to Lines 366-370 for the updated content. Appreciate your constructive suggestion.

Line 344. “All DEGs underwent k-means clustering analysis, categorizing 1,762 genes into six distinct expression clusters (116-467 genes per cluster; Figure 5B)”. Did this analysis have any significance beyond simple categorization? This categorization is not discussed further in the manuscript. The data shown in Figure 5A can be mentioned in the text without the need for a figure. If Figure 5B is not discussed, it is not necessary. Therefore, the entire Figure 5 is not necessary.

Response: Thanks for your advice and it’s very valuable for improving our paper. We have incorporated detailed explanations of the k-means clustering analyses for DEGs in both Section 3.4 (Results) and the Discussion section. Including (1) The expression pattern of specific DEGs clusters (lines 372-377), (2) the prediction of the synergistic effect between specific DEGs clusters and important terpene synthesis-related genes (lines 536-539, and lines 570-573).

In the Discussion section, the first paragraph (lines 454 to 461) is unnecessary. It is mentioned in the Introduction.

Response: Thank you for your comments. I have deleted the first paragraph of the discussion section.

Lines 97, 528, 531, 534: change bata to beta

Response: We are very sorry for our carelessness. We have changed bata to beta.

Round 2

Reviewer 1 Report

I would like to thank the authors for their thoughtful answers and their extensive efforts in addressing my and the other reviewers’ comments. In my opinion, this led to a much improved, and now publication-ready, manuscript. It was a pleasure to read through!

I only have one minor comment with respect to my previous comment 13, where I highlighted the following: "i.d. of the column is given in µM (micromolar) instead of µm (micrometer), please correct." I realize now that the value 1.8 refers to the column particle size, which should indeed be reported as µm (micrometer). Therefore, the text should be as follows: "dimensions: 100 mm × 2.1 mm; internal diameter particle size: 1.8 µ; Waters, Milford, United States)"

Author Response

Comments: I only have one minor comment with respect to my previous comment 13, where I highlighted the following: "i.d. of the column is given in µM (micromolar) instead of µm (micrometer), please correct." I realize now that the value 1.8 refers to the column particle size, which should indeed be reported as µm (micrometer). Therefore, the text should be as follows: "dimensions: 100 mm × 2.1 mm; internal diameter particle size: 1.8 µ; Waters, Milford, United States)"

Response: Thank you very much for your approval of the revisions to the manuscript. Additionally, I would like to express my sincere gratitude once again for your meticulous review of the revised draft. Per your suggestions, I have corrected the chromatographic column's dimensional specifications. Please refer to lines 150 to 151 for the updated information.